# A multicenter, randomized, parallel-group confirmatory study protocol to evaluate the efficacy of Soft Protector CPC, a novel oral mucosal protectant, in preventing oral mucositis and alleviating pain in patients with breast cancer

Kazuhiro Omori[1]*, Kohei Furukawa[2], Masatoshi Usubuchi[3¤], Tomofumi Hamada[4], Tadahiko Shien[5], Michihiro Yoshida[6], Yuki Nakatsuka[6], Katsuyuki Hotta[6], Soichiro Ibaragi[7], Shogo Takashiba[1]

1 Department of Pathophysiology-Periodontal Science, Faculty of Medicine, Dentistry and Pharmaceutical Sciences, Okayama University, Okayama, Japan, 2 Department of Dentistry and Oral Surgery, Shikoku Cancer Center, Ehime, Japan, 3 Department of Dentistry, Miyagi Cancer Center, Miyagi, Japan, 4 Department of Dentistry and Oral Surgery, Sagara Hospital, Kagoshima, Japan, 5 Department of Breast and Endocrine Surgery, Okayama University Hospital, Okayama, Japan, 6 Center for Innovative Clinical Medicine, Okayama University Hospital, Okayama, Japan, 7 Department of Oral and Maxillofacial Surgery, Faculty of Medicine, Dentistry and Pharmaceutical Sciences, Okayama University, Okayama, Japan

¤ Current Address: Department of Dentistry and Oral Surgery, Hakodate Medical Center, Hokkaido, Japan
* kazu@okayama-u.ac.jp

## Abstract

Oral mucositis is a frequent and debilitating adverse event observed in patients undergoing chemotherapy or radiotherapy. Current management strategies are limited in duration, require frequent application, and fail to address the mechanical irritation from teeth. A novel device, Soft Protector CPC, was developed to overcome these limitations. This multicenter, randomized, two-arm, open-label, confirmatory trial aims to evaluate the efficacy and safety of Soft Protector CPC in patients with breast cancer undergoing chemotherapy. A total of 154 participants will be randomly assigned in a 1:1 ratio to receive either oral care with Soft Protector CPC or oral care alone. The primary endpoint will be oral mucositis as assessed according to the Common Terminology Criteria for Adverse Events (CTCAE) v3.0 during the comparative treatment period. The secondary endpoints will include CTCAE v3.0 during the continuous treatment period, oral mucositis, pain (CTCAE v5.0), quality of life (Patient Reported Outcomes-CTCAE version 1.0 [PRO-CTCAE v1.0], the 15-item oral health questionnaire of the European Organization For Research And Treatment Of Cancer [EORTC QLQ-OH15], and the pain Numeric Rating Scale), onset and site of mucositis, completion of chemotherapy, use of rescue medications, technical feasibility, and patient preference. The safety endpoints will include adverse events, device malfunction, and laboratory tests. This trial is expected to establish the clinical utility of the Soft Protector CPC for the prevention and management of oral mucositis,

**Data availability statement:** No datasets were generated or analyzed during the current study. All relevant data from this study will be made available upon study completion.

**Funding:** This work was supported by the Japan Agency for Medical Research and Development (AMED; grant number JP25ck0106033). The funders had no role in study design, data collection and analysis, decision to publish, or preparation of the manuscript. San Medical Co., Ltd. provided the investigational material (Soft Protector CPC) for the study.

**Competing interests:** The authors received financial support from the Japan Agency for Medical Research and Development (AMED; grant number JP25ck0106033). This study was also supported by San Medical Co., Ltd., which provided the investigational material (Soft Protector CPC). This does not alter our adherence to PLOS ONE policies on sharing data and materials.

with the potential to improve the patients' quality of life and adherence to cancer therapy. This study was approved by the Clinical Research Review Board and registered with the Japan Registry of Clinical Trials, jRCTs062250005, on April 18, 2025.

## Introduction

Oral mucositis is a common and debilitating side effect in cancer patients undergoing chemotherapy or radiotherapy. It involves inflammation and ulceration of the oral mucosa, causing severe pain and affecting the patient's ability to eat, drink, and maintain oral hygiene. This condition can significantly reduce quality of life and may lead to treatment interruptions or dose reductions, potentially compromising cancer therapy outcomes [1–3]. Current management strategies for oral mucositis have limited effectiveness and effect duration. They often require frequent application and fail to address the mechanical irritation from teeth, which can exacerbate symptoms [4–6]. To overcome these limitations, novel approaches such as protective oral devices are being investigated to provide more comprehensive and longer-lasting relief for patients with chemotherapy-induced oral mucositis.

To address these issues, we developed a novel mucosal protectant, Soft Protector CPC, which has been approved as a medical device (Sun Medical Co., Ltd., approval no. 30500BZX00107000). The newly developed Soft Protector CPC differs from conventional therapies in that it is applied to teeth that are likely to mechanically irritate the oral mucosa, rather than being applied directly to ulcerated mucosal surfaces. By covering such teeth, the device is intended to reduce local mechanical stimulation associated with oral mucositis-related pain. Furthermore, Soft Protector CPC exerts an inhibitory effect on biofilm accumulation at its surface through the sustained release of cetylpyridinium chloride (CPC), an antimicrobial agent. Therefore, protective materials are expected to suppress the progression of dental caries and gingivitis/periodontitis.

A previous exploratory, randomized, controlled, two-arm, open-label trial was conducted at Okayama University Hospital without restrictions on cancer type (protocol registration no. jRCTs062220084). While patients with various cancers were enrolled, the majority were cases of breast cancer, in whom the trial demonstrated particularly favorable outcomes, including a notable pain-relieving effect on chemotherapy-induced oral mucositis.

Based on the favorable outcomes of our previous exploratory trial, which demonstrated promising pain-relieving effects of Soft Protector CPC, we designed this confirmatory clinical study to validate its preventive and therapeutic efficacy for oral mucositis in patients undergoing chemotherapy for breast cancer. In this multi-center, randomized controlled trial across four institutions, patients will be assigned to receive either oral care alone (primarily focused on moisturizing) or oral care with Soft Protector CPC. The incidence and severity of oral mucositis will be assessed primarily using the Common Terminology Criteria for Adverse Events (CTCAE) v3.0 for the confirmatory primary endpoint, while CTCAE v5.0 will be used for selected

secondary assessments of oral mucositis and oral pain. This study hypothesizes that the use of the Soft Protector CPC will significantly reduce the incidence and severity of oral mucositis compared to oral care alone.

## Materials and methods

This protocol was prepared in accordance with the SPIRIT 2025 guidelines. The completed SPIRIT checklist is provided as Supporting Information (S2 Checklist). The reporting of the trial results will follow the CONSORT guidelines.

### Objective, overall design, and framework

This randomized, controlled, two-arm, open-label confirmatory trial aimed to evaluate and confirm the efficacy and safety of Soft Protector CPC compared to oral care alone in preventing chemotherapy-induced oral mucositis and alleviating pain in patients with breast cancer. This trial was approved by the Clinical Research Review Board of the Okayama University (approval number: CRB24−024).

### Registration of the protocol

The Japan Registry of Clinical Trials (jRCT) registration number of this protocol is jRCTs062250005 (https://jrct.mhlw.go.jp/latest-detail/jRCTs062250005).

### Endpoints

**Assessment schedule.** The assessment schedule is summarized in Fig 1. Written informed consent will be obtained from all participants by principal investigator and co-investigators, and eligibility will be confirmed according to the inclusion and exclusion criteria. The participants will be registered after pre-registration examinations (screening tests) and final confirmation of eligibility. Enrolled participants will then be randomly assigned to either the intervention group (oral care with the application of the Soft Protector CPC, Fig 2) or the control group (oral care alone during the comparative period).

The study design is shown in Fig 3. During the randomized comparative period (1st course), the intervention group will receive oral care plus Soft Protector CPC, whereas the control group will receive standardized oral care alone. This randomized comparative period is the primary period for evaluating treatment efficacy and serves as the basis for confirmatory statistical analyses.

After completion of all assessments in the comparative period, all participants, regardless of their original allocation, will proceed to the continuous period (2nd course), during which Soft Protector CPC will be added to oral care in both groups. The data from this non-randomized continuous period will be used for exploratory purposes.

Each treatment period will generally be set to 21 days; however, if chemotherapy is administered in cycles, the duration of one cycle (14, 21, or 28 days) will be used. To minimize potential bias related to the timing of product application, the initiation of the test product will be synchronized with the first day of each chemotherapy cycle. In addition, the CTCAE and the 15-item oral health questionnaire of the European Organization For Research And Treatment Of Cancer [EORTC QLQ-OH15] used as endpoints are internationally recognized indicators that have also been adopted in clinical guidelines for the management of oral mucositis [7,8].

Study completion for an individual participant is defined as completion of the planned assessment at the end of the continuous period or, in cases of early discontinuation, completion of the final assessment performed at the time of discontinuation.

### Primary endpoint

The primary endpoint in this study will be oral mucositis as assessed by physicians according to the subcriteria from the CTCAE v3.0 during the comparative period. The primary endpoint is evaluated exclusively based on the data obtained

| Study Visit | V0 | V1 | V2 | V3 |
|---|---|---|---|---|
| **Enrollment & Allocation** | | **Post-allocation** | | |
| **Study Period** | | | **Comparative** | **Continuous** |
| **Timepoints: Day (d)** | -56d pre V1 | 0d | 14, 21, 28d[A] | 28, 42, 56d[A] |
| **Enrolment** | | | | |
| Informed consent | X | | | |
| Eligibility assessment | X | | | |
| Randomization | X | | | |
| **Interventions** | | | | |
| Intervention group | | X | ⟶ | ⟶ |
| Control group | | | X[B] | ⟶ |
| **Assessments** | | | | |
| CTCAE v3.0 sub-criteria | X | X | X | X |
| CTCAE v5.0 Oral mucositis & Pain | X | X | X | X |
| PRO-CTCAE v1.0 Items: #3a, 3b | X | X | X | X |
| EORTC-QLQ-OH15 Item:#35 | X | X | X | X |
| NRS | X | X | X | X |
| Blood test | X | | X | X |
| Oral exam | X | X | X | X |
| Oral photo | X | X | X | X |
| Panorama x-ray | X | | | |
| PRO | | X | ⟶ | ⟶ |
| Patient preference | | | X | X |
| Adverse events | | X | ⟶ | ⟶ |
| Device malfunctions | | X | ⟶ | ⟶ |

(A) Evaluations will be conducted in accordance with the chemotherapy administration schedule.

(B) Following completion of all assessments during the comparative period, the Soft Protector CPC will be placed in the oral cavity.

**Fig 1. SPIRIT schedule of enrolment, interventions, and assessments. (A)** Evaluations will be conducted in accordance with the chemotherapy administration schedule. **(B)** Following completion of all assessments during the comparative period, the Soft Protector CPC will be placed in the oral cavity. NRS; pain numerical rating scale, PRO; patient-reported outcome.

during the randomized comparative period. While similar assessment is performed during the continuous period, it is treated as exploratory endpoint. We will also evaluate the prevention of pain onset or the alleviation of pain. The endpoints are defined as follows:

• Prevention of onset: Absence of oral mucositis (Grade 0) at the affected site both at the initiation of study procedure (i.e., Day1 of the comparative period) and at the end of the evaluation period (i.e., the end of the comparative period).

(A)
(B)

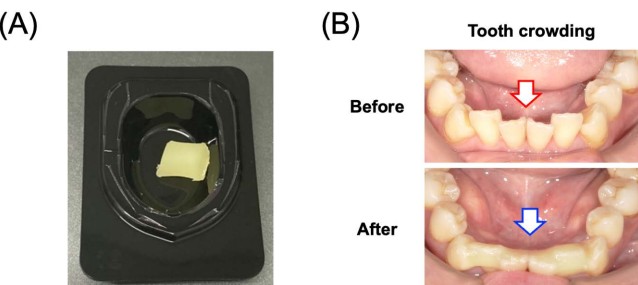

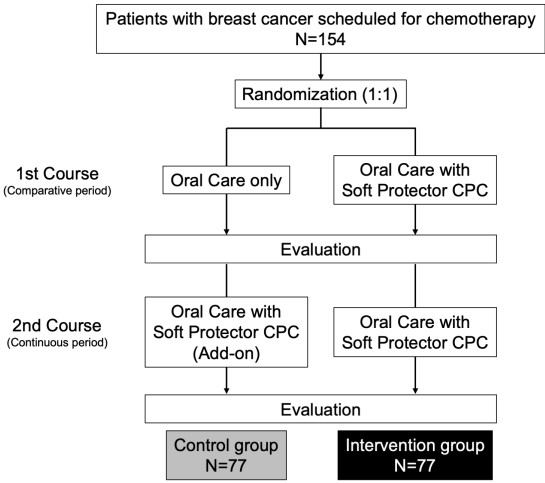

**Fig 2. Soft Protector CPC. (A)** Representative appearance of Soft Protector CPC. **(B)** Representative intraoral application of Soft Protector CPC to teeth that are likely to mechanically irritate the oral mucosa.

**Fig 3. Study design.** During the randomized comparative period (1st course), participants are assigned to either oral care plus Soft Protector CPC (intervention group) or oral care alone (control group). After completion of the comparative period, all participants proceed to the continuous period (2nd course), during which Soft Protector CPC is added to oral care in both groups. The continuous period is therefore a non-randomized add-on phase and is analyzed for exploratory purposes.

- Alleviating pain: in patients with oral mucositis of Grade 1, 2, or 3 at the initiation of the study procedure (i.e., Day1 of the comparative period), either a reduction of at least one grade or complete resolution of symptoms at the end of the evaluation period (i.e., the end of the comparative period).

- When the symptoms of grade 1 oral mucositis – according to the subcriteria from the Common Terminology Criteria for Adverse Events version 3 (CTCAE v3.0) – disappear, the patient's condition is defined as "absence of onset (for convenience, Grade 0)" as a supplemental definition specific to this study.

## Secondary endpoints

Secondary endpoints are pre-specified to support the interpretation of the primary endpoints; however, analyses of these endpoints will be considered exploratory, as the study is not powered for these outcomes.

1. Oral mucositis according to the subcriteria from the CTCAE v3.0 during the continuous period. The assessment will be conducted in the same manner as for the primary endpoint. The evaluation will be performed during the continuous period.

2. Oral mucositis and oral pain grading (CTCAE v5.0 and the Japanese version of the Medical Dictionary for Regulatory Activities version 25.1 [MedDRA/J v25.1]). Comparisons will be performed for both the comparative and continuous periods, assessing the changes before and after each period. The same procedure will be applied to the subsequent endpoints. The terms and definitions relevant to oral mucositis and oral pain are as follows:

   - Prevention of onset: absence of the respective condition (mucositis or pain) in the entire oral cavity before and after the evaluation period.

   - Alleviating pain: reduction in grade or absence of the respective condition for cases with grade 1, 2, or 3 before the evaluation period.

   - When the grade 1 oral pain—as defined according to the CTCAE v5.0 and the MedDRA/J v25.1—disappears, it is supplementarily defined as "absence of onset (for convenience, Grade 0)." However, Grade 0 is not defined for the symptoms of oral mucositis.

3. Quality of life (patient-reported outcomes). PRO-CTCAE v1.0 (items: #3a, #3b), EORTC QLQ-OH15 (item: #35), and pain numerical rating scale (NRS) at the affected site will be used. The NRS of the maximum pain during each treatment period will be derived from patient diaries. The NRS is scored from 0 to 10 (0 = no pain, 1–3 = mild pain, 4–6 = moderate pain, 7–10 = severe pain, with 10 being "the most intense pain ever experienced".

4. Occurrence and timing of oral mucositis: the presence of mucositis will be assessed across the entire oral cavity at each site (lips, tongue, buccal mucosa, palate, and other areas). Occurrence limited to the predefined affected sites will also be evaluated.

5. Subgroup analyses of pain alleviation or improvement of pain by the chemotherapy regimen.

6. Chemotherapy completion status.

7. Number of nonsteroidal anti-inflammatory drugs (NSAIDs; acetaminophen or loxoprofen sodium) taken (patient-reported outcome [PRO]).

8. Frequency of oral steroid ointment use as rescue therapy (PRO).

9. Use of Episil® oral liquid (PRO).

10. Technical feasibility (procedure-related assessments): spontaneous detachment, detachment during brushing or eating, and patient-initiated removal will be recorded.

11. Patient preference: patients will be asked whether they wish to continue the treatment with Soft Protector CPC.

**Safety endpoints**

1. Adverse events

2. Device malfunctions

3. Laboratory tests

## Eligibility criteria

**Inclusion criteria.** Participants must meet all the following criteria:

1. Patients with breast cancer undergoing chemotherapy, including molecular targeted therapy or immune checkpoint inhibitors at participating institutions.

2. Patients without oral candidiasis.

3. Patients with contact between the teeth and oral mucosa.

4. Patients aged ≥18 years at the time of consent.

5. An ECOG (Eastern Cooperative Oncology Group) Performance Status of 0 or 1.

6. Functionally capable of oral food intake.

**Exclusion criteria.** Participants who meet any of the following criteria will be excluded:

1. Patients scheduled to undergo head and neck radiotherapy and those with a history of head and neck radiotherapy.

2. Patients with fewer than 10 teeth.

3. Patients who use dentures.

4. Patients with known allergies to CPC or resin.

5. Pregnant or potentially pregnant patients.

6. Patients using or scheduled to use medical narcotics, opioid formulations, or oral steroid ointments (e.g., dexamethasone).

7. Patients scheduled for dental treatment.

8. Patients who are unable or unwilling to complete the required assessments under physician supervision.

9. Patients deemed unsuitable for participation for clinical or social reasons or due to conflicts of interest

## Randomization

Prior to study initiation, participants will be randomly assigned in a 1:1 ratio to the intervention and control groups. Randomization will be conducted using a stratified block method with study institution as the stratification factor.

To ensure comparability between the groups, the chemotherapies included in this study are limited to those frequently used at respective institutions. Furthermore, randomization stratified by institution is employed to maintain balance across different chemotherapies, thereby enhancing comparability between the groups.

Because this is an open-label trial, the detailed allocation algorithm will be documented in a separate specification that will remain confidential until completion of participant allocation.

## Treatment

**Intervention group.** Participants assigned to the intervention group will receive standardized oral care plus Soft Protector CPC during both the comparative and continuous periods.

Soft Protector CPC is a tooth-applied oral mucosal protective device and is not applied directly to the oral mucosa. The device is intended to reduce mechanical irritation to the oral mucosa by covering teeth that are considered likely to contact and injure the mucosa, such as teeth with crowding, sharp edges due to fracture, or marked attrition.

Before device application, the dentist will identify the target tooth or teeth on the basis of intraoral findings and the expected site of mucosal contact. If multiple candidate teeth are present, all relevant teeth will be treated and evaluated individually. Application will be performed only by the principal investigator or a co-investigator who is a licensed dentist and has been trained in the study procedure.

The device is supplied in a single-use blister package. After tooth surface preparation according to the study procedure, bonding material (Hybrid-Coat II, Sun Medical Co., Ltd.) will be used, and Soft Protector CPC will then be placed directly on the selected tooth surface using gloved fingers. Excess material will be removed, occlusion will be checked, and the material will be light-cured until complete hardening is achieved.

The device will remain in place throughout each treatment period (14, 21, or 28 days, according to the chemotherapy cycle). If chemotherapy is postponed, the device will be removed and reapplied on the rescheduled date. At the end of each treatment period, the device will be removed after outcome assessment. If device detachment occurs, the date, time, and presumed reason will be recorded. If the device deteriorates, causes difficulty in daily oral function, or if adverse events or discontinuation criteria arise, it will be removed for ethical and safety reasons, and end-of-treatment assessments will be performed.

**Control group.** During the comparative period, participants assigned to the control group will receive standardized oral care alone and will not receive Soft Protector CPC. After completion of the comparative period, they will enter the continuous period, during which Soft Protector CPC will be added to the same oral care regimen. Device application, removal, and discontinuation procedures during the continuous period will be identical to those used in the intervention group.

## Standardized oral care

Oral care is regarded as a background intervention in both groups and will be provided according to a standardized study procedure across participating centers. Oral care will primarily consist of oral moisturization, and routine supportive oral care judged appropriate for the study setting. Any rescue treatment or additional oral supportive care, including oral steroid ointment, Episil° oral liquid, and analgesic use, will be recorded throughout the study. Adherence to the assigned intervention will be monitored using physician records and participant-reported diaries.

## Safety monitoring

Given the low-risk nature of the intervention and the absence of serious adverse events in our previous exploratory study, no formal interim analysis is planned. All severe adverse events will be recorded using the EDC. The principal investigator will compile the oral adverse events reported by the site investigators and submit them to the Clinical Research Review Board of Okayama University.

## PRO

Patients will complete paper-based patient-reported outcome measures, including the weekly PRO-CTCAE v1.0 (#3a and #3b) and EORTC QLQ-OH15 (#35). In addition, patients will maintain a daily diary to record pain NRS at the affected site, device detachment, use of analgesics, and use of rescue treatments such as oral steroid ointment and Episil° oral liquid. These records will also be used to monitor adherence to the assigned intervention and background oral care during each study period.

## Data management

Data will be collected using the EDC system (UHCT ACReSS, Tokyo, Japan) by authorized personnel (principal investigator, sub-investigator, and research collaborators). Each participant will be assigned a study code, and de-identified data

will be used for analysis. All hard copies were de-identified and stored securely at Okayama University. The data will be retained for five years following the completion of the study and then destroyed.

## Sample size and statistical analyses

This study aims to confirm the superiority of oral care with Soft Protector CPC over oral care alone in preventing oral mucositis and alleviating the pain associated with this condition.

In a previous exploratory study, improvement of oral mucositis symptoms was evaluated according to the subcriteria from the CTCAE v3.0 after 15 days of Soft Protector CPC application. The results showed that 58.3% (7/12) achieved this endpoint with oral care plus Soft Protector CPC, compared with 0.0% (0/6) with oral care alone. Considering the limited sample size in the exploratory study and the differences in study design between the exploratory study and this trial, the proportion of patients achieving prevention of oral mucositis or alleviation of pain in this study was assumed to be 50.0% in the intervention group and 25.0% in the control group. These values were set as clinically meaningful and more conservative expected group differences compared to the findings of the previous exploratory study. Based on the aforementioned hypothesis, the effect size is Cohen's $h = 0.52$, corresponding to a medium effect size and suggesting a clinically meaningful difference. Using Pearson's chi-square test with a two-sided significance level of 5%, a sample of 154 patients (77 per group) was deemed necessary to achieve a statistical power of 90%. Considering the historical enrollment at the participating institutions, the planned sample size is feasible.

The primary endpoint will be the proportion of patients achieving prevention of the onset or alleviation of pain based on the subcriteria from the CTCAE v3.0 during the comparative period. The CTCAE grades at Day 1 and at the end of the comparative period will be descriptively summarized by treatment group, and a 95% confidence interval will be calculated by treatment group. The between-group difference in proportions and its two-sided 95% confidence interval will also be estimated. The superiority of Soft Protector CPC over the control treatment will be assessed using the Pearson's chi-square test with Yates' continuity correction at a two-tailed statistical significance level of 5%.

As a secondary analysis of the primary endpoint, we will estimate the risk difference in the proportion of patients between groups using a least-squares regression approach with robust standard errors [9]. The model includes center, baseline pain grade on Day 1 of the comparative period, and treatment group as fixed effects, with chemotherapy regimen treated as a random effect. From this model, we will report the point estimate for the treatment effect together with the 95% confidence interval and p-value. Furthermore, subset analyses will be conducted to strictly distinguish between preventive and alleviating effects according to the baseline pain grade; specifically, Grade 0 will be categorized as the onset prevention subset, while Grades 1, 2, and 3 will be categorized as the pain alleviation subset.

This confirmatory analysis is restricted to the comparative period, while the continuous period will be analyzed for exploratory purposes. All analyses will be performed on the full analysis set without imputation of missing values. For proportions, the denominator will be the total number of participants in the analysis set, including those with missing or unevaluable data.

## Interim analysis and monitoring

No interim analyses will be performed. Monitoring will be conducted by designated monitors in accordance with institutional SOPs. On-site (site visits and source data verification) and off-site (telephone, fax, and email) monitoring will be performed.

- Before initiation: Ethics approval, trial registration, document storage, and device management.

- Informed consent: verification of approved forms, correct documentation, and secure data storage

- Patient registration: Screening, eligibility, and allocation procedures.

 

- During treatment: verification of oral mucositis assessments, required tests, device performance, and data consistency. Monitoring will be performed at least once every six months.

- Discontinuation/serious events: verification of the discontinuation criteria met and regulatory reporting.

- Completion: Confirmation of proper document archiving, device management, and termination procedures.

## Dissemination

The results of this clinical trial will be presented at academic conferences, registered in a clinical trial registry (Japan Registry of Clinical Trials), and published in peer-reviewed journals.

## Ethics approval and current status of this trial

The study protocol (ver. 1.0) was performed in accordance with the Declaration of Helsinki and the Pharmaceutical Affairs Act of Japan and was approved by the Clinical Research Review Boards of Okayama University Hospital, Shikoku Cancer Center, Miyagi Cancer Center, and Sagara Hospital. The start date was April 18, 2025, and the first patient completed the registration on May 29, 2025. Four patients have been enrolled until now and more patients are being recruited from four institutions as of September 10, 2025. We plan to enroll 154 participants in the full analysis set. The limit of enrollment will be February 28, 2027, and the planned study will end on December 31, 2027. In addition, the study protocol (ver. 2.0) was approved on October 28, 2025.

## Discussion and conclusion

This confirmatory trial, developed based on findings from a previous exploratory study, was designed to generate high-quality evidence confirming the efficacy of Soft Protector CPC, a newly developed oral mucosal protectant, in preventing and managing chemotherapy-induced oral mucositis in patients with breast cancer. In an exploratory trial, a high incidence of oral mucositis was observed in patients undergoing chemotherapy for breast cancer. Previous reports have reported frequent occurrences of oral mucositis in patients receiving breast cancer chemotherapy regimens such as doxorubicin + cyclophosphamide (AC), docetaxel + cyclophosphamide (DC), and, more recently, datopotamab deruxtecan (Dato-DXd) [10–12]. The target population of the present study is patients with breast cancer. In addition, by focusing on both prevention and pain relief, and incorporating internationally recognized assessment tools (CTCAE, PRO-CTCAE, and EORTC QLQ-OH15) [7,8], this study addresses unmet clinical needs that are not fully covered by existing supportive care measures.

The strength of this study lies in its randomized multicenter design and emphasis on patient-reported outcomes, which are critical for evaluating both clinical efficacy and quality of life. Furthermore, this study comprises two periods: a comparison period and a continuous period. The participants of both groups will have the opportunity to attach the Soft Protector CPC during the continuous period. The inclusion of technical feasibility and patient preference assessments is expected to provide valuable insights into real-world applicability.

However, some limitations of this study should be acknowledged. First, because the random assignment is whether the Soft Protector CPC is attached, the trial is designed in an open-label setting, which may introduce bias in the evaluation. To minimize the influence of such bias, the primary endpoint was evaluated using the objective assessment criteria of oral mucositis, according to the subcriteria from the CTCAE v3.0. Furthermore, the study population was restricted to patients with breast cancer, and extrapolation to other cancer populations should be treated with caution. A small exploratory, randomized, controlled, two-arm, open-label study was conducted at Okayama University Hospital without restrictions of cancer type. As patient populations of different cancer types receive different chemotherapies and other treatments, we will prioritize improving the accuracy of statistical inference to confirm the effectiveness of this product using a reasonable sample size by specifying the cancer population.

Despite these limitations, the findings of this trial are expected to provide an evidence base for integrating the Soft Protector CPC into supportive care strategies, potentially improving quality of life and treatment adherence in patients with cancer.

## Supporting information

**S1. Checklist. PLOSOne human subjects research checklist.**
(PDF)

**S2. Checklist. SPIRIT 2025 editable checklist.**
(PDF)

**S3 File. Protocol Japanese.**
(PDF)

**S4 File. Protocol English translation.**
(PDF)

## Acknowledgments

We express our sincere gratitude to Dr. Masao Irie (Department of Biomaterials, Graduate School of Medicine, Dentistry and Pharmaceutical Sciences, Okayama University), Takashi Ito, Daisuke Uchida, and Jun Sakurai (Center for Innovative Clinical Medicine, Okayama University Hospital) for their valuable suggestions. We also thank Ms. Yuko Sugiura (Division of Dental Hygienists, Okayama University Hospital), Mr. Tomoaki Kuwano, Mr. Masaki Watanabe, and Ms. Naomi Kondo (Center for Innovative Clinical Medicine, Okayama University Hospital) for their excellent technical support. We are grateful to Ms. Eriko Yokoyama and Ms. Fumiko Tanabe (Department of Pathophysiology-Periodontal Science, Graduate School of Medicine, Dentistry, and Pharmaceutical Sciences, Okayama University) for their secretarial support. In addition, we wish to thank Mr. Tatsuya Ori, Mr. Narimichi Honda, Mr. Yuya Yamamoto, Ms. Akari Shimozono-Yamamoto, Mr. Yoshihiro Miura, Ms. Hiromi Managi, Ms. Emi Ohta (Sun Medical Co., Ltd.), Mr. Tomohisa Takagi, and Mr. Yuji Futamura (J Morita Corp.) for their support in the development of the Soft Protector CPC.

We would like to thank Editage (www.editage.com) for the English language editing.

## Author contributions

**Conceptualization:** Kazuhiro Omori.

**Formal analysis:** Michihiro Yoshida, Yuki Nakatsuka.

**Funding acquisition:** Kazuhiro Omori.

**Investigation:** Kazuhiro Omori, Kohei Furukawa, Masatoshi Usubuchi, Tomofumi Hamada.

**Methodology:** Kazuhiro Omori, Kohei Furukawa, Masatoshi Usubuchi, Tomofumi Hamada, Tadahiko Shien, Michihiro Yoshida, Yuki Nakatsuka.

**Project administration:** Kazuhiro Omori.

**Supervision:** Katsuyuki Hotta, Soichiro Ibaragi, Shogo Takashiba.

**Validation:** Michihiro Yoshida, Yuki Nakatsuka.

**Writing – original draft:** Kazuhiro Omori.

**Writing – review & editing:** Kazuhiro Omori, Kohei Furukawa, Masatoshi Usubuchi, Tomofumi Hamada, Tadahiko Shien, Michihiro Yoshida, Yuki Nakatsuka, Katsuyuki Hotta, Soichiro Ibaragi, Shogo Takashiba.

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
