## [Decision Letter · Decision Letter 0]

24 Feb 2026

PONE-D-25-56784A multicenter, randomized, parallel-group confirmatory study protocol to evaluate the efficacy of Soft Protector CPC, a novel oral mucosal protectant, in preventing oral mucositis and alleviating pain in patients with breast cancerPLOS One

Dear Dr. Omori,

Thank you for submitting your manuscript to PLOS ONE. After careful consideration, we feel that it has merit but does not fully meet PLOS ONE’s publication criteria as it currently stands. Therefore, we invite you to submit a revised version of the manuscript that addresses the points raised during the review process.

**The reviewers made specific comments please address them, apologies for taking this long with the review process.**

We look forward to receiving your revised manuscript.

Kind regards,

Benjamin Benzon, Ph.D., M.D.

Academic Editor

PLOS One

Journal Requirements:

This research is supported by the Japan Agency for Medical Development (AMED) under grant number JP25ck0106033.

5. Thank you for stating the following in the Competing Interests/Financial Disclosure section:

The authors received financial support from the Japan Agency for Medical Research and Development (grant number JP25ck0106033). This study supported by San Medical Co., Ltd., for providing the investigational material (Soft Protector CPC).

We note that you received funding from a commercial source: San Medical Co., Ltd.,

6. Please ensure that you refer to Figure 1 in your text as, if accepted, production will need this reference to link the reader to the figure.

7. We note that Figure 2 in your submission contain copyrighted images. All PLOS content is published under the Creative Commons Attribution License (CC BY 4.0), which means that the manuscript, images, and Supporting Information files will be freely available online, and any third party is permitted to access, download, copy, distribute, and use these materials in any way, even commercially, with proper attribution. For more information, see our copyright guidelines: http://journals.plos.org/plosone/s/licenses-and-copyright.

8. We note that there is identifying data in the Supporting Information file <20251028_S1_Protocol_Japanese.docx, 20251028_S2_Protocol_English translation.docx>. Due to the inclusion of these potentially identifying data, we have removed this file from your file inventory. Prior to sharing human research participant data, authors should consult with an ethics committee to ensure data are shared in accordance with participant consent and all applicable local laws.

-Location data

Reviewers' comments:

Reviewer's Responses to Questions

**Comments to the Author**

1. Does the manuscript provide a valid rationale for the proposed study, with clearly identified and justified research questions?

Reviewer #1: Yes

Reviewer #2: Yes

Reviewer #3: Yes

2. Is the protocol technically sound and planned in a manner that will lead to a meaningful outcome and allow testing the stated hypotheses?

Reviewer #1: Yes

Reviewer #2: Partly

Reviewer #3: Partly

3. Is the methodology feasible and described in sufficient detail to allow the work to be replicable?

Reviewer #1: Yes

Reviewer #2: Yes

Reviewer #3: No

4. Have the authors described where all data underlying the findings will be made available when the study is complete?

Reviewer #1: Yes

Reviewer #2: Yes

Reviewer #3: Yes

5. Is the manuscript presented in an intelligible fashion and written in standard English?

Reviewer #1: Yes

Reviewer #2: Yes

Reviewer #3: Yes

6. Review Comments to the Author

You may also provide optional suggestions and comments to authors that they might find helpful in planning their study.

Reviewer #1: As the statistical reviewer I will focus on methods and reporting

Major

1) The power calculation is necessarily arbitrary. report as cohen's d for clarity, the effect size. is ICC expected since this is a multi centre trial? is there no chance of drop outs? that is not accounted in the power calculations.

2) there is no mention of the consort statement and its relevance here.

3) i'd advise analyses to follow a regression framework so effect sizes and uncertainty can be easily quantified and reported - moving away a bit from p-values, they shouldn't be the focus.

4) what characteristics will be reported for each arm, on which balanced will be evaluated? what will the authors do if the randomisation has not ensured balanced across all covariates considering the small sample? and will missing data be dealt with (perhaps multiple imputation as a sensitiivty analysis)

5) again going back to expected heterogeniety, or not, across centres, the authors need to consider how this affects the study. in the power calculation but also in the analyses. in a regression framework one would expect a mixed effects model with a random intercept for centre.

6) there are numerous secondary outcomes. analyses for these are likely to be underpowered and authors need to be clear these analyses will be described as exploratory.

7) any interventions (medications) need to be monitored and pehaps used in the regression model if there is no complete balance.

Minor

1) is there a plan for safety to be monitored statistically? this relates to the fact there are no interim analyses planed.

Reviewer #2: GENERAL ASSESSMENT

This manuscript describes a multicenter, randomized, open-label controlled trial protocol evaluating the efficacy of the Soft Protector CPC for the prevention or alleviation of chemotherapy-induced oral mucositis in patients with breast cancer. The topic is clinically relevant, and the randomized design and inclusion of both clinician-assessed and patient-reported outcomes are strengths. However, the protocol raises substantial methodological concerns related to the definition of the primary endpoint, its alignment with the two study periods, and the justification of the sample size and statistical analysis, all of which affect the interpretability and confirmatory strength of the study.

MAJOR COMMENTS:

1. Primary endpoint definition and clinical relevance

The primary endpoint is defined as “the proportion of patients achieving prevention of the onset or alleviation of pain during the comparative period” and is planned to be analyzed as a binary outcome using a Pearson chi-square test. This definition raises important concerns. It combines two conceptually distinct phenomena: prevention of oral mucositis onset, which applies to the entire study population, and alleviation of pain, which is relevant only to the subset of patients who develop symptoms. Collapsing these constructs into a single binary outcome obscures clinical interpretation and makes patient classification unclear.

In addition, the protocol does not specify how “alleviation of pain” is operationally defined (e.g., magnitude of change, timing, or applicable patient subset), leaving this component open to subjective interpretation and inconsistent application. These issues are particularly relevant given that oral mucositis in breast cancer chemotherapy is of moderate incidence: published data suggest rates of approximately 20–40%, with severe cases (CTCAE grade ≥3) in fewer than 10% of patients (e.g., Elting et al. 2003 DOI: 10.1002/cncr.11671; Sonis, 2004 DOI: 10.1038/nrc1318; Lalla et al. 2014 DOI: 10.1002/cncr.28592). As many participants may never develop mucositis or pain, inclusion of pain alleviation within the primary endpoint risks inflating apparent success without clearly reflecting a preventive effect. The primary endpoint should therefore be redefined with clear operational criteria and aligned with the preventive nature of the study population.

2. Relationship between endpoints and study periods

The protocol distinguishes between a randomized “comparative period” and a subsequent “continuous period” during which all participants receive Soft Protector CPC. While this design may be ethically motivated, its analytical implications are not sufficiently clarified. The comparative period is the only phase that allows a randomized comparison and must therefore underpin all primary efficacy conclusions. Outcomes assessed during the continuous period, when no control group remains, cannot support causal inference and should be considered descriptive or exploratory.

However, the protocol does not clearly restrict the primary endpoint to the comparative period nor explicitly define the continuous period as exploratory, creating a risk of overinterpretation of non-randomized results. This ambiguity is further compounded by the unclear endpoint definitions, particularly if symptom alleviation is emphasized beyond the randomized comparison. The authors should clearly delineate the analytical roles of the two study periods and explicitly limit confirmatory conclusions to the comparative period.

3. Statistical analysis inconsistencies

The sample size calculation is based on the primary endpoint described above and on a Pearson chi-square test comparing proportions between groups. The validity of this approach depends on a clearly defined binary outcome with a well-specified underlying probability structure. Given the composite and ambiguously defined nature of the primary endpoint, it is unclear how the assumed response rates used for the power calculation were derived or justified. In particular, the protocol appears to assume a relatively high success rate in the intervention group (>50%), which is difficult to reconcile with the moderate incidence of chemotherapy-induced oral mucositis in breast cancer patients unless the endpoint definition is overly broad.

Moreover, the chi-square–based power calculation implicitly assumes a uniform binary outcome assessed over a comparable observation window for all participants. This assumption is questionable in light of the heterogeneous occurrence of mucositis, the mixture of preventive and symptom-based components within the endpoint, and the variability in chemotherapy cycle length across patients. Consequently, the sample size justification lacks a transparent clinical and statistical foundation.

The protocol further specifies Pearson’s chi-square test as the sole inferential method for evaluating the primary endpoint while simultaneously stating an intention to demonstrate superiority of the intervention. The use of superiority language implies a directional hypothesis, which is conceptually inconsistent with a non-directional testing framework unless explicitly justified. Additionally, given the anticipated event rates and the composite nature of the endpoint, some contingency table cells may have small expected counts, potentially violating chi-square test assumptions. The authors should therefore clarify the intended hypothesis structure, ensure consistency between hypothesis, statistical test, and power calculation, and provide a clear justification for the chosen analytical approach.

MINOR COMMENTS:

1. Different CTCAE versions (v3.0 and v5.0) are referenced; a single version should be specified and consistently applied.

2. The procedures for screening, enrollment, and allocation concealment are insufficiently described and should be clarified.

3. The standard of oral care provided to the control group should be more precisely defined to ensure consistency across centers.

4. The duration of participation per patient and the definition of study completion should be explicitly stated.

CONCLUSION

This protocol addresses an important clinical question and has the potential to generate useful evidence. However, substantial clarification is required regarding the definition of the primary endpoint, its relationship to the study periods, and the justification of the statistical analysis and sample size. Addressing these issues will significantly strengthen the methodological rigor and interpretability of the trial.

Reviewer #3: All detailed comments are provided in the attached PDF document of the submitted manuscript.

The main concern relates to the description of all interventions planned in this study. Currently, the descriptions are vague and insufficiently detailed, and they should be more clearly and comprehensively reported. In particular, the protocol should explicitly describe how the tested device will be applied in the clinical setting, in accordance with the TIDieR checklist, to ensure transparency, reproducibility, and scientific rigor.

7. PLOS authors have the option to publish the peer review history of their article (what does this mean?). If published, this will include your full peer review and any attached files.

Reviewer #1: No

Reviewer #2: **Yes:** Darko Kero

Reviewer #3: No

---

## [Author Response · Author response to Decision Letter 1]

24 Apr 2026

Journal Requirements:

We thank the editor for this important comment. In accordance with the PLOS ONE style requirements, we have revised the manuscript formatting, including file naming, to ensure compliance with the journal’s guidelines and templates.

We thank the editor for this important comment. In accordance with the journal’s instruction, we have provided the following complete Data Availability Statement in the submission form: “No datasets were generated or analyzed during the current study. All relevant data from this study will be made available upon study completion.

We thank the editor for this important comment. In accordance with the editor’s suggestion, we have checked the funding information carefully and corrected the grant number(s) in the “Funding Information” section to ensure consistency with the “Financial Disclosure” section.

4. Thank you for stating the following financial disclosure: This research is supported by the Japan Agency for Medical Development (AMED) under grant number JP25ck0106033. Please state what role the funders took in the study. If the funders had no role, please state: "The funders had no role in study design, data collection and analysis, decision to publish, or preparation of the manuscript." If this statement is not correct you must amend it as needed. Please include this amended Role of Funder statement in your cover letter; we will change the online submission form on your behalf.

We thank the editor for this important comment. In accordance with the editor’s instruction, we have added the amended Role of Funder statement to the cover letter.

5. Thank you for stating the following in the Competing Interests/Financial Disclosure section: The authors received financial support from the Japan Agency for Medical Research and Development (grant number JP25ck0106033). This study supported by San Medical Co., Ltd., for providing the investigational material (Soft Protector CPC). We note that you received funding from a commercial source: San Medical Co., Ltd., Please provide an amended Competing Interests Statement that explicitly states this commercial funder, along with any other relevant declarations relating to employment, consultancy, patents, products in development, marketed products, etc. Within this Competing Interests Statement, please confirm that this does not alter your adherence to all PLOS ONE policies on sharing data and materials by including the following statement: "This does not alter our adherence to PLOS ONE policies on sharing data and materials.” (as detailed online in our guide for authors http://journals.plos.org/plosone/s/competing-interests). If there are restrictions on sharing of data and/or materials, please state these. Please note that we cannot proceed with consideration of your article until this information has been declared. Please include your amended Competing Interests Statement within your cover letter. We will change the online submission form on your behalf.

We thank the editor for this important comment. In accordance with the editor’s instruction, we have amended the Competing Interests Statement to explicitly disclose the commercial support from San Medical Co., Ltd. and included the revised statement in the cover letter. We have also confirmed that this does not alter our adherence to all PLOS ONE policies on sharing data and materials.

6. Please ensure that you refer to Figure 1 in your text as, if accepted, production will need this reference to link the reader to the figure.

We thank the editor for this comment. In accordance with the editor’s suggestion, we have added a reference to Figure 1 in the main text.

7. We note that Figure 2 in your submission contain copyrighted images. All PLOS content is published under the Creative Commons Attribution License (CC BY 4.0), which means that the manuscript, images, and Supporting Information files will be freely available online, and any third party is permitted to access, download, copy, distribute, and use these materials in any way, even commercially, with proper attribution. For more information, see our copyright guidelines: http://journals.plos.org/plosone/s/licenses-and-copyright. We require you to either (1) present written permission from the copyright holder to publish these figures specifically under the CC BY 4.0 license, or (2) remove the figures from your submission:

We thank the editor for pointing this out. Following the editor’s instruction, we removed the copyrighted exterior photograph(s) from Figure 2 and revised the submitted figure to ensure compliance with PLOS ONE’s copyright and licensing requirements.

8. We note that there is identifying data in the Supporting Information file <20251028_S1_Protocol_Japanese.docx, 20251028_S2_Protocol_English translation.docx>. Due to the inclusion of these potentially identifying data, we have removed this file from your file inventory. Prior to sharing human research participant data, authors should consult with an ethics committee to ensure data are shared in accordance with participant consent and all applicable local laws.

-Location data

We thank the editor for pointing this out. Following the editor’s instruction, we carefully reviewed the Supporting Information files, removed all potentially identifiable personal information, and prepared fully anonymized versions in accordance with participant privacy protection requirements.

We thank the editor for pointing this out. Following the editor’s instruction, we added the captions for the Supporting Information files at the end of the manuscript and revised the corresponding in-text citations to ensure consistency.

We thank the editor for this helpful comment. In accordance with the editor’s suggestion, we evaluated the publications recommended by the reviewers and incorporated the relevant references into the revised manuscript where appropriate.

Reviewer #1: As the statistical reviewer I will focus on methods and reporting

Major

1) The power calculation is necessarily arbitrary. report as cohen's d for clarity, the effect size. is ICC expected since this is a multi centre trial? is there no chance of drop outs? that is not accounted in the power calculations.

Thank you for your careful review and the insightful comments. We provide our responses to each point below.

i) The power calculation is necessarily arbitrary.

Response:

As described in the “Sample size and statistical analyses” section of the Materials and Methods, in our previous study, 58.3% (7/12) of patients in the group using Soft Protector CPC alongside oral care achieved the primary endpoint, compared to 0.0% (0/6) in the oral care-only group. Given the limited sample size of the previous exploratory study, the proportions for the power calculation of this trial were set at 50.0% for the intervention group and 25.0% for the control group. These values were established as clinically meaningful and more conservative expected group differences compared to the findings of the previous exploratory study.

Therefore, the sample size for this study was determined conservatively based on the results of the exploratory research and is not based on arbitrary assumptions. We have added the following description to the manuscript to clarify this point:

Section: Sample size and statistical analyses

These values were set as clinically meaningful and more conservative expected group differences compared to the findings of the previous exploratory study.

ii) Report as Cohen's d for clarity, the effect size.

Response:

Since the primary endpoint of this study is a binary variable, we calculated the effect size using Cohen’s h. Based on the aforementioned hypothesis, the effect size is h = 0.52, which corresponds to a medium effect size and is considered a clinically meaningful difference. We have added this information to the manuscript as follows:

Section: Sample size and statistical analyses

Based on the aforementioned hypothesis, the effect size is Cohen’ s h = 0.52, corresponding to a medium effect size and suggesting a clinically meaningful difference.

iii) Is ICC expected since this is a multi-centre trial?

Response:

The number of institutions participating in this study is small, and it is anticipated that there will be differences in the number of feasible participants between institutions. Therefore, we considered it difficult to appropriately adjust for the Intraclass Correlation Coefficient (ICC) during the sample size design phase. Furthermore, since the types of chemotherapy regimens for patients within an institution are not limited to a small number, the similarity between such patients is unlikely to be high. Consequently, the ICC is expected to be close to 0, and we believe that even if an adjustment were made, its impact would be minimal. We plan to evaluate the influence of inter-institutional differences during the analysis stage (see response to Reviewer#1 Major3)4)5)7) and Section Sample size and statistical analyses).

iv) Is there no chance of dropouts? That is not accounted in the power calculations.

Response:

In the previously exploratory study, no dropouts occurred, regardless of cancer type. It should be noted that during the middle of the evaluation period, some male cancer patients expressed difficulty with frequent visits due to their business work commitments. In contrast, female breast cancer patients were cooperative with the study and did not raise such concerns. Based on these experiences, we have designed this trial to limit the timing of product application and evaluation strictly to start date of the chemotherapy administration cycle, thereby eliminating unimportant hospital visits. We believe this makes dropouts for such reasons less likely to occur.

Furthermore, since the comparative evaluation period covers only one cycle of cancer treatment, mid-term dropout is unlikely. Additionally, not a single adverse event with a causal relationship to this product was observed in the exploratory study. For these reasons, we consider the dropout rate to be extremely low and do not account for its impact in the sample size design or statistical analysis.

However, in response to your comment, we have performed the following additional consideration as a precaution. The required sample size for this study is set at 154 cases with a power of 90%. Under a generally accepted power of 80%, the required sample size would be 116 cases. Therefore, although we consider such a situation unlikely, sufficient power would be maintained even if a dropout rate of approximately 24.6% (38 cases) were to occur.

2) there is no mention of the consort statement and its relevance here.

Response：

Thank you for this valuable comment. We have clarified the reporting framework by specifying the relevant reporting guidelines in the revised manuscript. Specifically, we have added a statement indicating that this study protocol was developed in accordance with the SPIRIT2025 statement, which is the appropriate reporting guideline for randomized clinical trial protocols. We have also noted that the trial results will be reported in accordance with the CONSORT guidelines. We hav

---

## [Decision Letter · Decision Letter 1]

19 May 2026

A multicenter, randomized, parallel-group confirmatory study protocol to evaluate the efficacy of Soft Protector CPC, a novel oral mucosal protectant, in preventing oral mucositis and alleviating pain in patients with breast cancer

PONE-D-25-56784R1

Dear Dr. Omori,

We’re pleased to inform you that your manuscript has been judged scientifically suitable for publication and will be formally accepted for publication once it meets all outstanding technical requirements.

Kind regards,

Yu Uneno

Academic Editor

PLOS One

Additional Editor Comments (optional):

Reviewers' comments:

Reviewer's Responses to Questions

**Comments to the Author**

1. Does the manuscript provide a valid rationale for the proposed study, with clearly identified and justified research questions?

Reviewer #1: Yes

2. Is the protocol technically sound and planned in a manner that will lead to a meaningful outcome and allow testing the stated hypotheses?

Reviewer #1: Yes

3. Is the methodology feasible and described in sufficient detail to allow the work to be replicable?

Reviewer #1: Yes

4. Have the authors described where all data underlying the findings will be made available when the study is complete?

Reviewer #1: Yes

5. Is the manuscript presented in an intelligible fashion and written in standard English?

Reviewer #1: Yes

6. Review Comments to the Author

You may also provide optional suggestions and comments to authors that they might find helpful in planning their study.

Reviewer #1: I am satisfied with all the authors' responses and the resulting changes to the paper. I have nothing else to add.

7. PLOS authors have the option to publish the peer review history of their article (what does this mean?). If published, this will include your full peer review and any attached files.

Reviewer #1: No

---

## [Editor Report · Acceptance letter]

PONE-D-25-56784R1

PLOS One

Dear Dr. Omori,

I'm pleased to inform you that your manuscript has been deemed suitable for publication in PLOS One. Congratulations! Your manuscript is now being handed over to our production team.

Kind regards,

on behalf of

Dr. Yu Uneno

Academic Editor

PLOS One